# Gamma Knife stereotactic radiotherapy combined with tislelizumab as later-line therapy in pMMR/MSS/MSI-L metastatic colorectal cancer: a phase II trial analysis

Yiran Zhang[1,2†], Hanyang Guan[1†], Shijin Liu[1†], Haoquan Li[1], Zili Bian[1], Jiashuai He[1], Zhan Zhao[1], Shenghui Qiu[1], Tianmu Mo[1], Xiangwei Zhang[1], Zuyang Chen[1], Hui Ding[1], Xiaoxu Zhao[1], Liang Wang[3]*, Yunlong Pan[1,4]*, Jinghua Pan[1]*

[1]Department of General Surgery, The First Affiliated Hospital of Jinan University, Guangzhou, China; [2]Department of Body Gamma Knife, The First Affiliated Hospital of Jinan University, Guangzhou, China; [3]Department of Oncology, The First Affiliated Hospital of Jinan University, Guangzhou, China; [4]MOE Key Laboratory of Tumor Molecular Biology and Key Laboratory of Functional Protein Research of Guangdong Higher Education Institutes, Institute of Life and Health Engineering, Jinan University, Guangzhou, China

*For correspondence:
wangliang@jnu.edu.cn (LW);
tpanyl@jnu.edu.cn (YP);
huajanve@foxmail.com (JP)

†These authors contributed equally to this work

Competing interest: The authors declare that no competing interests exist.

## eLife Assessment

This **valuable** study found Gamma Knife SBRT combined with tislelizumab offers a safe and powerful later-line option for pMMR/MSS/MSI-L metastatic CRC patients who were unresponsive to the first and second-line chemotherapy. The authors implemented a well-structured experimental protocol and provide **convincing** evidence to substantiate the conclusions. This work would be of broad interest to oncologists working on colorectal cancer.

## Abstract

**Background:** An immunosuppressive tumor microenvironment limits the efficacy of immunotherapy, thus patients with MSS and pMMR mCRC often face great challenges.

**Methods:** In this phase II trial, patients received Gamma Knife SBRT combined with Tislelizumab. Biomarker analysis was performed pre- and post-treatment.

**Results:** From November 2022 to July 2024, 1of 20 patients achieved CR, 13 of 20 patients achieved PR, 6 achieved SD. mPFS was 10.7 months (95% CI, 6.4-15.0). With no grade 4 events noted, common adverse events included nausea (65%), anemia (55%), and fatigue (45%). RNA sequencing indicated enhanced immune infiltration in PR patients. For patients with pMMR/MSS/MSI-L mCRC who had not responded to first and second-line therapies, the combo of Gamma Knife SBRT and tislelizumab showed high efficacy and reasonable safety. Significant post-radiotherapy improvements in the tumor's immunosuppressive microenvironment, including lower fibrosis, normalizing of tumor vasculature, and activation of the PD-1/PD-L1 checkpoint pathway were revealed by biomarker analysis.

**Conclusions:** These results imply that patients with pMMR/MSS/MSI-L mCRC who were unresponsive to the first and second-line chemotherapy, Gamma Knife SBRT with tislelizumab provides a safe and powerful later-line treatment alternative.

**Funding:** This research was supported by the Clinical Frontier Technology Program of the First Affiliated Hospital of Jinan University (No. JNU1AF-CFTP-2022-a01223), the National Natural

Science Foundation of China (82204436), Natural Science Foundation of Guangdong Province (2024A1515030010, 2022A1515011695), Science and Technology Projects in Guangzhou (2024A03J0825).
**Clinical trial number:** ChiCTR2200066117.

## Introduction

Colorectal cancer continues to represent a significant threat to life. As reported in the 2020 Global Cancer Statistics, colorectal cancer accounts for 10% of all cancer cases, ranking third in incidence, while its mortality rate is 9.4%, second only to lung cancer (*Sung et al., 2021*; *Siegel et al., 2020*). Especially, 20% of newly diagnosed colorectal cancer patients present show metastases, and 40% have recurrence and metastases after local treatment (*Biller and Schrag, 2021*). The FOLFOX/FOLFIRI chemotherapy regimen, which comprises oxaliplatin, 5-fluorouracil, and irinotecan, is the mainstay of clinical treatment for metastatic colorectal cancer (mCRC). For patients harboring wild-type RAS and BRAF, the addition of the epidermal growth factor receptor (EGFR) inhibitor cetuximab is recommended (*Diagnosis, 2019*; *Benson et al., 2021*). For patients with RAS mutations, the anti-angiogenic agent bevacizumab is advised. Nevertheless, RAS-mutant patients exhibit poorer prognoses and shorter survival times compared to their wild-type counterparts (*Cox et al., 2014*; *Modest et al., 2016*). The efficacy of chemotherapy in combination with targeted therapy remains suboptimal (*Modest et al., 2016*).

The development of immune checkpoint inhibitors (ICIs) transforms cancer immunotherapy (*Asaoka et al., 2015*). Particularly, CRCs with mismatch repair deficiency and high microsatellite instability show a strong response to ICIs (*Ganesh et al., 2019*). But most CRC cases are either microsatellite-stable/low microsatellite instability (MSS/MSI-L) or mismatch repair-profile (pMMR), which reduces the efficacy of immunotherapy in a significant number of mCRC patients (*Ganesh et al., 2019*). Chemotherapeutic agents can cause immunogenic cell death in tumors, thus coordinating with ICIs improves antitumor efficacy (*Yi et al., 2022*). Additionally, antiangiogenic therapies targeting VEGFR facilitate the normalization of tumor vasculature and promote immune cell infiltration, subsequently amplifying immune-mediated tumor eradication (*Limagne et al., 2016*). Clinical studies, however, have revealed that combining mFOLFOX6 or other chemotherapy regimens with anti-VEGF, anti-EGFR, and ICIs does not produce better clinical outcomes in mCRC (*Hamid et al., 2024*; *Antoniotti et al., 2022*). Consequently, identifying alternative strategies to augment the efficacy of immunotherapy remains a pivotal objective in the field of cancer immunotherapy in pMMR/MSS/MSI-L mCRC.

Stereotactic body radiation therapy (SBRT) effectively targets and eradicates tumor cells with high-dose radiation (*Papiez et al., 2003*). Although traditional radiotherapy is sometimes linked with immunosuppressive effects (*Mac Manus et al., 1997*), SBRT's exact targeting can expose tumor neoantigens, mobilize and activate immune cells, increase their infiltration into the tumor, and improve the tumor immune microenvironment (*Singh et al., 2017*; *Choi et al., 2019*). The Gamma Knife is a principal modality in SBRT, employing gamma rays generated by cobalt-60 to deliver a single, high-dose focused irradiation to the target lesion. The Gamma Knife provides several benefits over conventional radiotherapy, including exact stereotactic targeting, increased delivery dose to the lesion, prevention of accelerated repopulation of tumor cells, and better local control rates of tumors (*Morinaga et al., 2016*). Our team first observed a case with pMMR-type rectal cancer who exhibited local recurrence and distant metastasis following first-line and second-line chemotherapy combined with targeted therapy (*Liu et al., 2022*). After undergoing Gamma Knife SBRT followed by tislelizumab treatment, intrahepatic metastatic lesions were reduced and stabilized. The patient showed a partial response (PR) with notable reduction of recurrent lesions in the rectal wall and stabilization of intrahepatic metastases, so extending the progression-free survival (PFS) exceeded beyond 3 months (*Liu et al., 2022*). These findings suggest that Gamma Knife SBRT might improve ICB sensitivity in mCRC.

The results of a phase II clinical trial assessing the combination of Gamma Knife SBRT combined with tislelizumab as a later-line therapy in patients with pMMR/MSS/MSI-L mCRC are presented in this report together with safety and efficacy. NanoString assay for transcriptome analysis was employed to elucidate changes in the tumor immune microenvironment during the combined treatment, offering insights into the therapeutic potential and mechanistic underpinnings of this integrated approach.

**Table 1.** Baseline demographic and clinical characteristics.

| Characteristics | Patients (n=20) |
|---|---|
| Age, years, median (IQR), n (%) | 60 (56–65) |
| <60 | 8 (40%) |
| ≥60 | 12 (60%) |
| Sex, n (%) | |
| Male | 15 (75%) |
| Female | 5 (25%) |
| ECOG performance status, n (%) | |
| 0 | 12 (60%) |
| 1 | 8 (40%) |
| Primary tumor location, n (%) | |
| Left colon and rectum | 17 (85%) |
| Right colon | 3 (15%) |
| Number of metastatic organs*, n (%) | |
| 1 | 14 (70%) |
| ≥2 | 6 (30%) |
| Metastatic organ, n (%) | |
| Liver | 14 (70%) |
| Lung | 7 (35%) |
| Lymph node | 2 (10%) |
| Other | 3 (15%) |
| Ras mutation type, n (%) | |
| KRAS | 5 (25%) |
| NRAS | 6 (30%) |
| Other | 9 (45%) |
| PD-L1 expression, CPS, n (%) | |
| CPS ≤1 | 12 (60%) |
| CPS＞1 | 6 (30%) |
| Unknown | 2 (10%) |
| TMB (mut/Mb), median (IQR), n (%) | 4.62 (3.08–8.97) |
| TMB <5 | 4 (20%) |
| TMB ≥5, ≤10 | 3 (15%) |
| TMB >10 | 1 (5%) |
| Unknown | 12 (60%) |

Abbreviations: CPS, combined positive score; ECOG, Eastern Cooperative Oncology Group; IQR, interquartile range; TMB, tumor mutation burden.
*Multiple answers allowed.

## Results

### Patients

In this clinical trial, 20 patients with pMMR/MSS/MSI-L tumors refractory to first- or second-line treatment were enrolled. The cohort comprised 15 males and 5 females, with ages ranging from 47 to

77 years. Predominantly, the primary tumors were located in the left colon and rectum (17/20, 85%), with the liver being the most common site of metastasis, followed by the lung (3/20) (*Table 1*). Flow-chart of the therapeutic regimen and flow diagram of enrolled participants in the study were shown in *Figure 1A* and *Figure 1B*.

Molecular profiling revealed RAS mutations in 11 patients (55%), with 5 exhibiting KRAS mutations and 6 presenting NRAS mutations. PD-L1 expression was assessed in 18 patients, and 12 (60%) patients had a combined positive score (CPS)≤1. Tumor mutation burden (TMB) data were available for 8 patients, with a median TMB of 4.62 mutations/Mb (IQR 3.08–8.97). Notably, only 1 patient exhibited a TMB>10 mutations/Mb (*Table 1*).

## Efficacy

In our cohort of 20 patients meeting inclusion criteria,1(5%) achieved clinical response (CR) ,12 (60%) achieved a partial response (PR) and 6 (30%) maintained stable disease (SD), resulting in a robust disease control rate (DCR) of 95% (*Table 2*). Patients with liver metastases achieved a 92.9% DCR, and patients with metastases in non-liver locations notably achieved a remarkable 100% DCR. Only one patient with liver metastases experienced disease progression (PD) (*Figure 2A*). As of the data cutoff date, seven patients remained on maintenance treatment, and one patient underwent surgery due to PD (*Figure 2A*). Remarkably, three patients refractory to first-line treatment responded to SBRT combined with tislelizumab, achieving rapid regression to NED status, with durations ranging from 6 to 18 months before progression. Encouragingly, one patient remains in a state of NED, under ongoing monitoring (*Figure 2A*).

Most patients exhibited favorable survival outcomes throughout the treatment (*Figure 2B*), and median progression-free survival (mPFS) was 10.7 months (95% CI, 6.4, 15.0) (*Figure 2C*). Additionally, a comparative survival analysis included 23 patients who underwent first- and second-line treatment and Gamma Knife SBRT without immunotherapy, revealing an mPFS of 6.7 months (95% CI, 5.6, 7.0). This data showed that Gamma Knife SBRT combined with tislelizumab as later-line treatment prolonged PFS in mCRC (log-rank test = 5.638, p=0.0176) (*Figure 2D*). These findings suggest that Gamma Knife SBRT combined with tislelizumab can effectively inhibit mCRC progression.

In our analysis of clinical characteristics and patient prognosis, we observed that the mPFS of patients with wild-type RAS expression was significantly longer (14 months) compared to those with KRAS (7.3 months) and NRAS (5 months) mutations (*Figure 2—figure supplement 1A*). These results are consistent with prior studies, indicating that RAS mutations are associated with increased malignancy, diminished therapeutic response, and poorer overall prognosis in colorectal cancer. However, statistical analysis did not reveal a significant difference between the groups (log-rank test = 4.278, p=0.118), which may be attributed to the limited sample size and other factors inherent to the study design. Additionally, we assessed the PFS of patients with simple liver metastases versus those with metastases to other organs (*Figure 2—figure supplement 1B*). Notably, the mPFS of patients with liver metastases (13 months) was longer than that of those with metastases at other sites (8.5 months). This difference could potentially be explained by the efficacy of localized Gamma Knife treatment in managing liver metastases. Despite this observation, statistical comparison between the two groups did not yield a significant difference (Log-rank test = 0.081, p=0.776). Given these limitations, further investigation with larger sample sizes is essential to elucidate the factors contributing to these trends and to confirm the clinical significance of these findings.

In light of the abscopal effect of radiotherapy, we extended our observations beyond the lesions directly targeted by stereotactic radiotherapy to include nonirradiated lesions (*Figure 2—figure supplement 1C*). Imaging examinations revealed significant tumor regression in both the irradiated target lesions (*Figure 2E and F*) and the nonirradiated lesions (*Figure 2E and G*) following Gamma Knife SBRT combined with tislelizumab. These findings suggest that SBRT not only impacts the irradiated lesions but also sensitizes distant metastatic sites for ICBs through the abscopal effect, thereby enhancing the systemic antitumor response when combined with immunotherapy.

## Safety

All 20 enrolled patients received the assigned treatment regimen, with safety assessments conducted every three treatment cycles. Treatment-related adverse events and immune-related adverse events are summarized in *Table 3*. Predominantly, patients experienced mild to moderate adverse events,

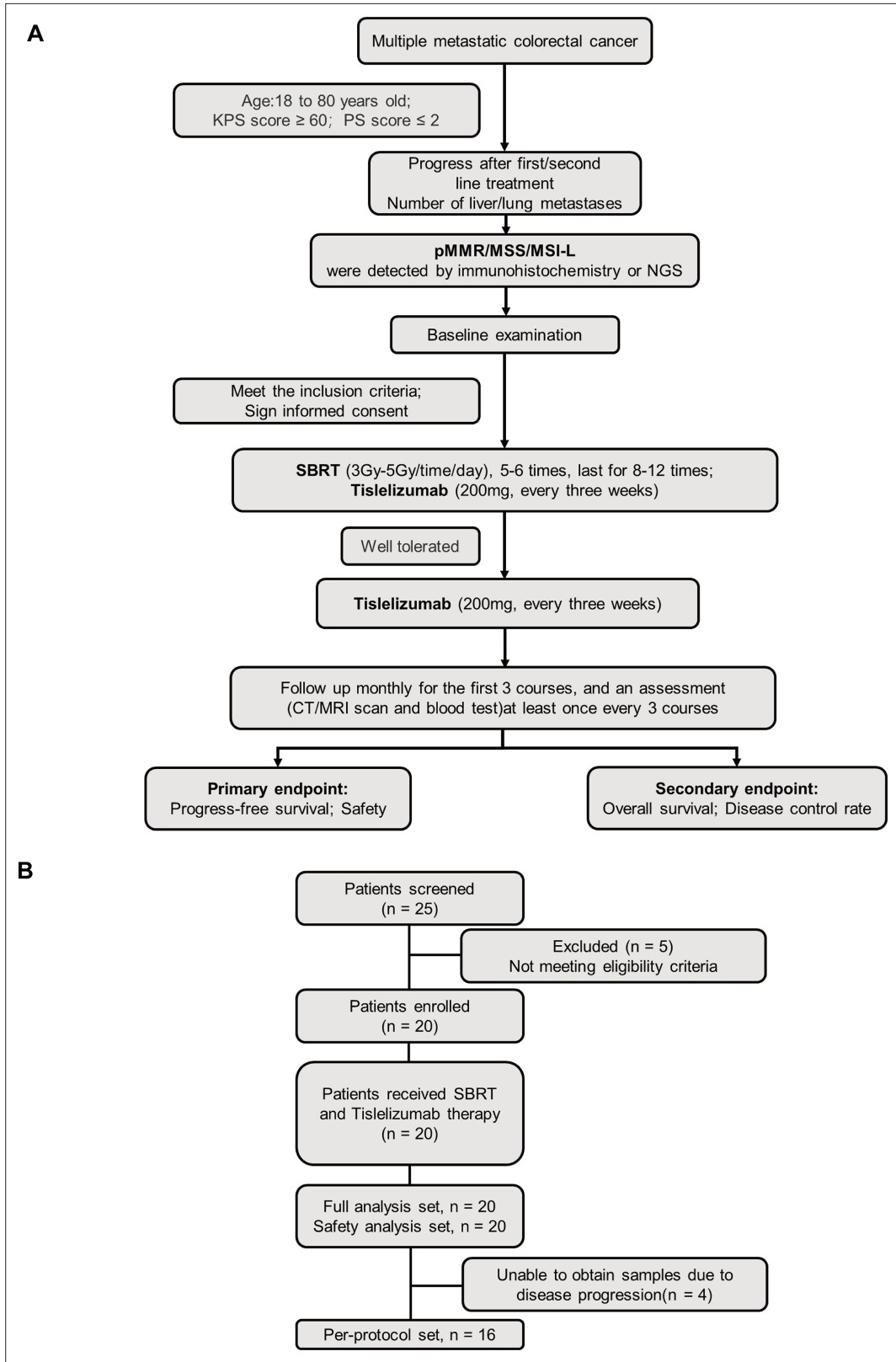

**Figure 1.** Clinical trial flowchart. (**A**) Flowchart of therapeutic regimen. (**B**) Flow diagram of participants in the study.

**Table 2.** Efficacy outcomes.

| | All patients (N=20) | Liver metastasis (N=14) | Other metastasis (N=6) |
|---|---|---|---|
| Best overall response | | | |
| Complete response (CR), n (%) | 1 (5%) | 0 (0%) | 1 (17%) |
| Partial response (PR), n (%) | 12 (60%) | 8 (57%) | 4 (66%) |
| Stable disease (SD), n (%) | 6 (30%) | 5 (36%) | 1 (17%) |
| Progressive disease (PD), n (%) | 1 (5%) | 1 (7%) | 0 (0%) |
| ORR, n (%, 95% CI) | 13 (65%, 40.8–84.6%) | 8 (57.1%, 28.9–82.3%) | 5 (83.3%, 35.9–99.6%) |
| DCR, n (%, 95% CI) | 19 (95%, 75.1–99.9%) | 13 (92.9%, 66.1–99.8%) | 6 (100%, 54.1–100%) |

Abbreviations: ORR, objective response rate; DCR, disease control rate.

with the most common being nausea (65%), anemia (55%), electrolyte disturbances (55%), fatigue (45%), and anorexia (35%). Notably, only two patients experienced grade 3 events of increased blood bilirubin, while no grade 4 adverse events were reported throughout the study period.

## Identification of differentially expressed genes between responder and nonresponse groups

To elucidate the impact of the tumor immune microenvironment on combination therapy outcomes, we employed NanoString assay for transcriptome analysis of tumor samples obtained from 16 enrolled patients before and after treatment, totaling 32 samples (*Figure 3A*). Patients were stratified into responder (PR) and nonresponder (non-PR) groups based on treatment outcomes. Gene expression differential analysis between pre- and posttreatment samples within each group identified significant alterations, as detailed in the Source data and illustrated in *Figure 3B*.

Our findings highlighted notable upregulation of key genes involved in antigen presentation (CD40, TNFSF18, TNFSF4), immune checkpoint modulation (PDCD1LG2, CD274, IDO1, VTCN1), and T cell activation pathways (TNFRSF9, CD28, ICOS, CD40LG, CD2, GZMK, ENTPD1, ITGAE) in the responder group. Additionally, a diverse array of chemokine family genes (IL2, IL4, IL17A, CCR2, CCL22) showed enhanced expression in the PR group (*Figure 3B*). Furthermore, immune cell abundance analysis based on 11 predefined immune cell types revealed significantly elevated levels in the PR group compared to non-PR. These included T cells, B cells, mast cells, macrophages, dendritic cells (DCs), cytotoxic cells, NK CD56 cells, CD8 T cells, CD45 cells, Th1 cells, and NK cells (*Figure 3C*). This heightened immune activation in responders encompassed robust antigen presentation, T cell activation, and co-stimulation processes crucial for effective immune-mediated tumor control.

To investigate the impact of Gamma Knife treatment on patient outcomes, we conducted a differential expression analysis of pre- and posttreatment samples from both responder (PR) and nonresponder (non-PR) groups (*Figure 3—figure supplement 1A*). The results revealed limited gene expression alterations following treatment. However, intersecting the differentially expressed genes from both groups identified NOS2 as a gene that was consistently upregulated posttreatment in both cohorts (*Figure 3—figure supplement 1B*). Although NOS2's increased expression may be associated with Gamma Knife treatment, its potential role in colorectal cancer treatment remains underexplored, warranting further investigation to elucidate the underlying mechanisms.

Additionally, to identify genes linked to treatment efficacy, we performed univariate and multivariate Cox regression analyses on the expression levels of 289 genes from sequencing data in relation to PD. Unfortunately, due to the limited sample size and sequencing depth, neither analysis yielded statistically significant results. The full Cox regression analysis of these genes is provided in *Supplementary file 1*.

Following the combination of stereotactic radiotherapy and immunotherapy, a striking reduction in liver metastasis target lesions was observed in two patients compared to baseline. To elucidate these findings, we conducted CD8 and PD-L1 immunohistochemical staining on liver metastasis biopsy specimens from some patients pre- and posttreatment (*Figure 3D and E*). The analysis revealed

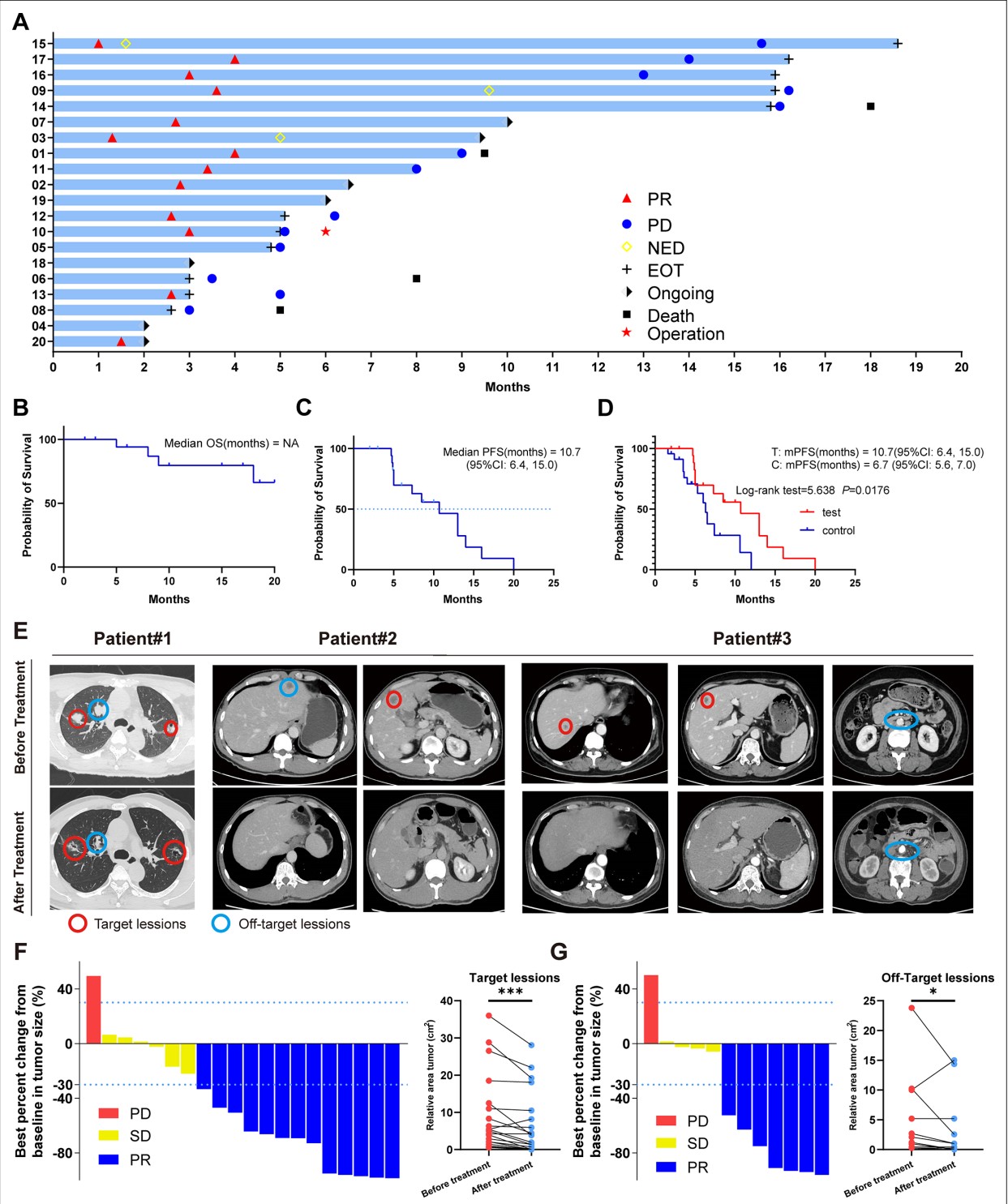

**Figure 2.** Clinical trial results. (**A**) Swimmer plots of patients. (**B**) Kaplan-Meier curves of OS for the per-protocol set (N=20). (**C**) Kaplan-Meier curves of progression-free survival (PFS) for the per-protocol set (N=20). (**D**) Kaplan-Meier curves of PFS for didn't receive immunotherapy set (control group) (N=23) and per-protocol set (test group) (N=20). (**E**) Radiological response from patient. (**F**) Waterfall plot of best percent change from baseline in patient target lesion (N=20). (**G**) Waterfall plot of best percent change from baseline in patient off-target lesion (N=12).

The online version of this article includes the following figure supplement(s) for figure 2:

**Figure supplement 1.** Supplementary clinical trial results.

**Table 3.** Treatment-emergent adverse events (TEAEs) since the initiation of protocol-specified treatment.

| TEAEs, n (%) | Patient (N=20) | | | | |
|---|---|---|---|---|---|
| | Grade 1 | Grade 2 | Grade 3 | Grade 4 | Any grade |
| Anemia | 9 (45%) | 2 (10%) | 0 | 0 | 11 (55%) |
| Neutropenia | 1 (5%) | 0 | 0 | 0 | 1 (5%) |
| Nausea | 10 (50%) | 3 (15%) | 0 | 0 | 13 (65%) |
| Poor appetite | 3 (15%) | 4 (20%) | 0 | 0 | 7 (35%) |
| Electrolyte disturbance | 7 (35%) | 2 (10%) | 0 | 0 | 11 (55%) |
| Hand-foot syndrome | 0 | 0 | 0 | 0 | 0 |
| Leukocytopenia | 2 (10%) | 0 | 0 | 0 | 2 (10%) |
| Aspartate transaminase increased | 2 (10%) | 2 (10%) | 0 | 0 | 4 (20%) |
| Lipase increased | 0 | 0 | 0 | 0 | 0 |
| Proteinuria | 0 | 0 | 0 | 0 | 0 |
| Thrombocytopenia | 2 (10%) | 2 (10%) | 0 | 0 | 4 (20%) |
| Vomiting | 1 (5%) | 3 (15%) | 0 | 0 | 4 (20%) |
| Hypothyroidism | 0 | 0 | 0 | 0 | 0 |
| Triglycerides increased | 0 | 0 | 0 | 0 | 0 |
| Fatigue | 6 (30%) | 3 (15%) | 0 | 0 | 9 (45%) |
| Blood bilirubin increased | 0 | 0 | 2 (10%) | 0 | 2 (10%) |
| Alanine transaminase increased | 2 (10%) | 1 (5%) | 0 | 0 | 3 (15%) |
| Peripheral neurotoxicity | 0 | 0 | 0 | 0 | 0 |
| Hoarseness | 0 | 0 | 0 | 0 | 0 |
| Rash | 4 (20%) | 0 | 0 | 0 | 4 (20%) |
| Thyroiditis | 0 (0%) | 0 | 0 | 0 | 0 |
| Diarrhea | 1 (5%) | 0 | 0 | 0 | 1 (5%) |
| Troponin increased | 0 | 0 | 0 | 0 | 0 |
| Fever | 0 | 0 | 0 | 0 | 0 |
| Alkaline phosphatase increased | 0 | 0 | 0 | 0 | 0 |
| Amylase increased | 0 | 0 | 0 | 0 | 0 |
| Hypertension | 0 | 0 | 0 | 0 | 0 |

increased infiltration of T cells and improved immune microenvironment following treatment, aligning with our prior analytical findings.

## Additional immune signatures analysis in predicting tumor response

We conducted gene expression analysis based on 12 predefined gene sets associated with immunotherapy and prognosis (*Figure 4A*). Notably, samples from the responder group exhibited higher expression of immune activation-related genes compared to the nonresponder group, including effector T cells (T-eff), T cell-inflamed, IFN-γ, cytotoxic, cytolytic activity score (CYT), chemokines, angiogenesis (AG), APC co-stimulation (APC co-sti), inflammation promoting (Inflam-pro), T cell co-stimulation (T cell co-sti), parainflammation (parainflam), and tumor-infiltrating lymphocytes (TIL) (*Figure 4A*).

Further compared to the related-signature score, we found the responders had higher APC and T cell co-stimulation signature scores compared with the nonresponder group (*Figure 4B*). Moreover, the responders had higher T cell-inflamed, inflammation-promoting, and parainflammation signature

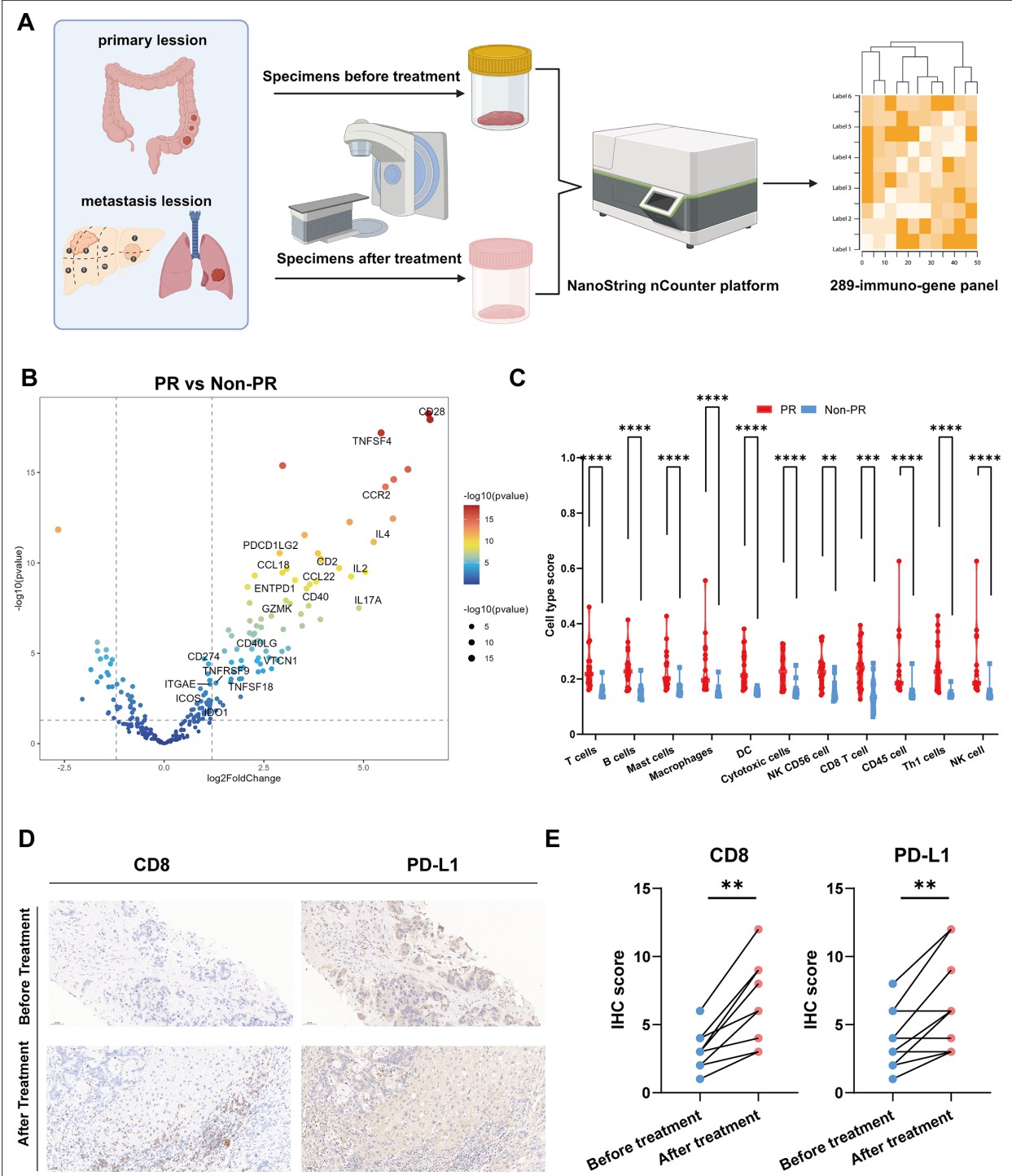

**Figure 3.** Differentially expressed genes analysis. (**A**) Specimens collection flowchart. (**B**) Transcriptome analysis on differential expression genes between responders (PR) (n=9) and nonresponders (Non-PR) (n=7). DESeq2 was provided to perform differential expression testing. (**C**) The abundance of predefined 12 immune cell composition before and after treatment between responders (PR) (n=9) and nonresponders (Non-PR) (n=7). The Wilcoxon test was used to determine the statistical significance between subgroups. (**D and E**) Representative CD8 and PD-L1 immunohistochemistry (IHC) staining of before and after treatment specimens of the patient.

The online version of this article includes the following figure supplement(s) for figure 3:

**Figure supplement 1.** Supplementary differentially expressed genes analysis.

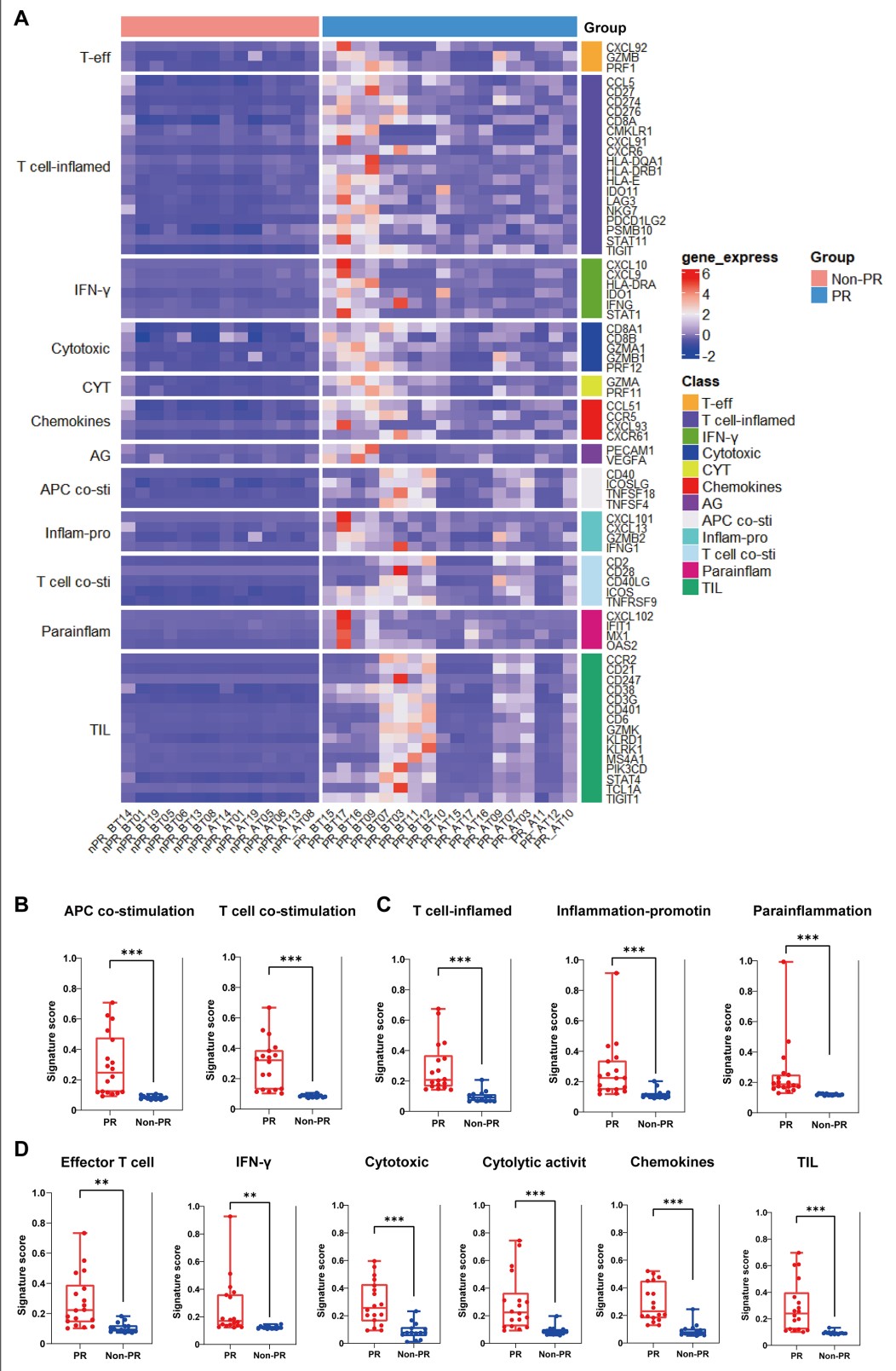

**Figure 4.** Additional immune signatures analysis. (**A**) The expression of 12 gene sets previously reported to be associated with response to immunotherapy and prognosis between responders (PR) (n=18) and nonresponders (Non-PR) (n=14). (**B–D**) 11 gene sets of prognostic value were differentially expressed between responders (PR) (n=18) and nonresponders (Non-PR) (n=14). Box plots are indicated in terms of minima, maxima, center, bounds

*Figure 4 continued on next page*

*Figure 4 continued*

of box and whiskers (interquartile range value), and percentile in the style of Tukey. The Wilcoxon test was used to determine the statistical significance between subgroups.

The online version of this article includes the following figure supplement(s) for figure 4:

**Figure supplement 1.** Gene Ontology (GO) enrichment and Kyoto Encyclopedia of Genes and Genomes (KEGG) pathways analysis of differential expression genes.

scores compared with the nonresponder group (*Figure 4C*). Additionally, increased expression of effector T cell, cytotoxicity, IFN-γ production, and cytolytic activity and TIL signature scores compared with the nonresponder group (*Figure 4D*).

Further functional insights into differential gene expression between responder and nonresponder groups were gained through Gene Ontology (GO) enrichment and Kyoto Encyclopedia of Genes and Genomes (KEGG) pathway analyses. GO analysis highlighted enrichment in cytokine and chemokine receptor activities, alongside increased T cell and leukocyte proliferation and activation levels in responders (*Figure 4—figure supplement 1A*). Correspondingly, KEGG analysis underscored enrichment in pathways involving antigen processing and presentation, T cell receptor signaling, chemokine interactions, and cytokine signaling (*Figure 4—figure supplement 1B*). Notably, these results indicated that responders after the combination of Gamma Knife SBRT and tislelizumab treatment will enhance tumor antigen presentation and T cell-mediated immune response in pMMR/MSS/MSI-L mCRC.

## Analysis on differential expression genes before and after treatment in the responders

To unravel the mechanisms driving tumor regression in the responder cohort, we conducted comprehensive gene expression analysis before and after treatment, focusing on seven gene groups known for their potential inhibitory effects on immunotherapy. Posttreatment analysis revealed significant reductions in exhausted T cells, Th2 cells, and Treg cells, indicative of a favorable shift away from a suppressive immune microenvironment (*Figure 5A*). Tumor resistance mechanisms, such as fibrosis and angiogenesis, play pivotal roles in limiting therapeutic efficacy (*Herzog et al., 2023*; *Kopecka et al., 2021*). Initially, the evaluation of immunotherapy-related gene groups in partial responders versus nonresponders highlighted significantly higher angiogenesis scores in the former, albeit with no significant difference (*Figure 5B*). Recognizing potential biases from pooling samples pre- and posttreatment, we conducted separate analyses within the responder group, expanding our gene set to include fibrosis-related genes. The findings underscored substantial inhibition of both angiogenesis and fibrosis within the tumor microenvironment following SBRT, targeted therapy, and immunotherapy (*Figure 5C*). We also analyzed tumor samples from select patients before and after treatment, employing immunohistochemical staining for CD31, a marker of vascular survival (*Figure 5D*). The results demonstrated a significant reduction in CD31 expression following Gamma Knife irradiation, further supporting the notion that Gamma Knife treatment, when combined with immunotherapy, can effectively inhibit tumor angiogenesis (*Figure 5E*).

Further stratified analysis of nonresponder samples before and after treatment revealed no significant alterations in the expression levels of immunosuppression-related or angiogenesis/fibrosis-related gene sets (*Figure 5—figure supplement 1*). These insights illuminate critical pathways through which combined therapies modulate the immune landscape and enhance treatment responses in pMMR/MSS/MSI-L mCRC.

## Discussion

By successfully reaching its main endpoint, this phase II trial shows that combined Gamma Knife SBRT with tislelizumab greatly increases PFS in pMMR/MSS/MSI-L mCRC, resistant to first- and second-line therapies. For this patient population, the combo treatment has shown both safety and tolerability. By overcoming resistance to first treatment plans, our study presents a creative therapy approach for those unresponsive to conventional treatments that offers a suitable therapeutic option improving clinical outcomes.

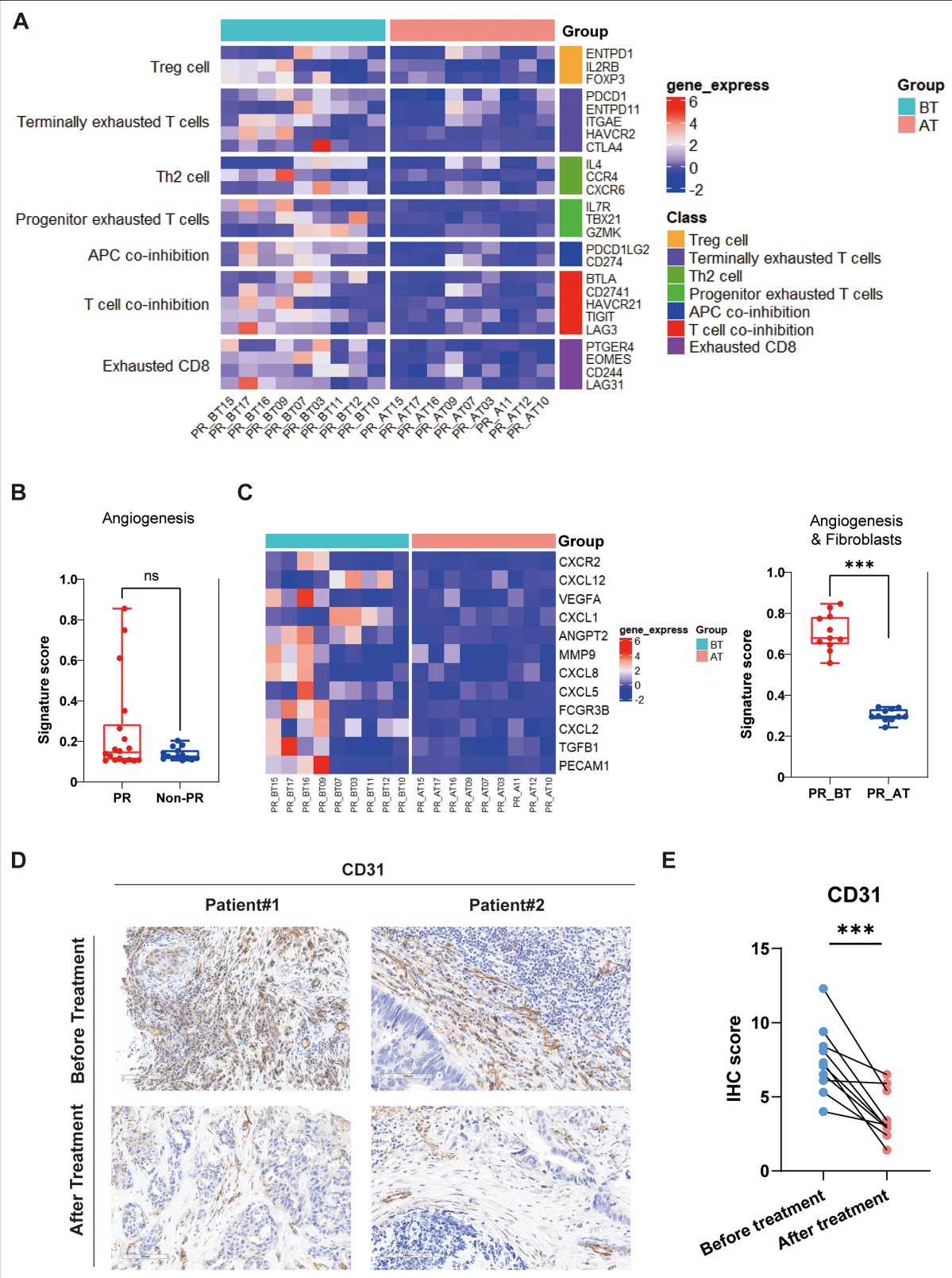

**Figure 5.** Comparison of responders before and after treatment. (**A**) The expression of seven gene sets previously reported to be associated with response to immunosuppressive between before treatment (n=9) and after treatment (n=9) in the responders (PR). (**B**) The expression of angiogenesis sets between responders (PR) (n=18) and nonresponders (Non-PR) (n=14). (**C**) The expression of angiogenesis and fibroblasts sets between before treatment (n=9) and after treatment (n=9) in the responders. Box plots are indicated in terms of minima, maxima, center, bounds of box and whiskers

*Figure 5 continued on next page*

*Figure 5 continued*

(interquartile range value), and percentile in the style of Tukey. The Wilcoxon test was used to determine the statistical significance between subgroups. (**D and E**) Representative CD31 immunohistochemistry (IHC) staining of before and after treatment specimens of patients.

The online version of this article includes the following figure supplement(s) for figure 5:

**Figure supplement 1.** Comparison of nonresponders before and after treatment.

Among the several cancers including nasopharyngeal carcinoma, esophageal cancer, liver cancer, and lung cancer, tislelizumab, a new PD-1 inhibitor, has been shown especially therapeutic efficacy. Combining tislelizumab with chemotherapy has essentially extended PFS in patients across these cancers (*Yang et al., 2023*; *Wang et al., 2021*; *Shen et al., 2022*; *Qin et al., 2023*). While immunotherapy has proven beneficial for some patients, mCRC presents unique challenges. Particularly in patients with MSS/pMMR tumors, which are marked by low immunogenicity and great resistance to immunotherapy, tumor cells in mCRC often evade immune detection and destruction (*Zhao et al., 2022*). By directly targeting and destroying tumor cells, Gamma Knife SBRT presents a potential solution by releasing a significant volume of tumor neoantigens and improving tumor immunogenicity, thus optimizing and maximizing the efficacy of subsequent immunotherapy (*Kievit et al., 2023*). Furthermore, demonstrated to extend survival in non-small cell lung cancer with patients with brain metastases is the combination of Gamma Knife SBRT and immunotherapy (*Cho et al., 2020*). Still underreported, though, is the possibility of Gamma Knife SBRT coupled with ICIs to improve the response in pMMR/MSS/MSI-L CRC.

In our clinical observations, a notable therapeutic effect was achieved in a patient treated with combined SBRT and immunotherapy. We hypothesize that the addition of tislelizumab following SBRT could extend PFS compared to either modality alone. Tumor microenvironment post-radiotherapy showed significant changes revealed by sequencing analysis of tumor samples both before and after combined treatment. More precisely, the microenvironment transitioned from an immunosuppressive, angiogenesis- and fibrosis-promoting state to an immune-enhanced, angiogenesis- and fibrosis-attenuated state. Comparatively to nonresponders, responders expressed genes linked to antigen presentation, tumor inflammation, and immune-mediated tumor killing more strongly. Further showing the activation of several signaling pathways associated with tumor cell death, including NF-κB, TNF, and JAK-STAT pathways, was enrichment analysis. Furthermore, immunotherapy targets such as PD-L1 showed an elevation, which supports the possibility of efficient later immunotherapy. These findings substantiate our hypothesis that patients with MSS-type mCRC resistant to first-line treatment could benefit significantly from the combination of stereotactic radiotherapy and immunotherapy, with enhanced immunogenicity and a more favorable tumor microenvironment facilitating improved therapeutic outcomes.

This trial has restrictions even if its outcomes show promise. First of all, our results could be biased as a single-arm study devoid of a control group. Second, the limited sample size and single-center design of the study lower its statistical power; hence, more robust conclusions depend on bigger studies. Furthermore, even though general survival (OS) was examined, the follow-up duration was insufficient to establish a reliable median OS. Subgroup analysis stratified by mutation type and metastatic site did not yield statistically significant prognostic results, likely due to limitations in both the available patient data and the sequencing depth. Given the importance of identifying reliable biomarkers for predicting the efficacy of Gamma Knife treatment, we attempted to address this gap by establishing a Cox proportional hazards regression model. This model aimed to correlate specific genetic and clinical features with treatment outcomes. Despite these efforts, no substantial or statistically significant findings emerged from the analysis, indicating that additional data and more comprehensive modeling may be required to uncover potential predictors of treatment response. To address these limitations, a multicenter, randomized controlled trial with a larger cohort and extended follow-up period is essential. This will provide a more comprehensive evaluation of the efficacy and safety of combining Gamma Knife SBRT and tislelizumab as a later-line therapy in pMMR/MSS/MSI-L mCRC patients.

Ultimately, for patients with pMMR/MSS/MSI-L mCRC who were unresponsive to first-line therapy regimens, the combination of Gamma Knife SBRT with tislelizumab demonstrated a high DCR and a manageable safety profile. Significant post-radiotherapy improvements in the tumor's suppressive immune microenvironment, reduced fibrosis, normalized tumor vasculature, and activation of

the PD-1/PD-L1 checkpoint pathway revealed by biomarker analyses, so improving the efficacy of immunotherapy.

## Methods

### Study design and participants

This single-arm, phase II trial was conducted at the First Affiliated Hospital of Jinan University to assess the antitumor efficacy and safety of a combined regimen consisting of SBRT and tislelizumab in patients with pMMR/MSS/MSI-L-type mCRC. The study is registered with Chinese Clinical Trial Registry (identifier: ChiCTR2200066117). Eligible patients, aged 18–75 years, had confirmed mCRC. MSS and RAS mutation statuses were determined through gene sequencing, while clinical staging was based on imaging examinations and intraoperative findings. A total of 20 patients were enrolled in the study, with all providing written informed consent. Detailed inclusion and exclusion criteria are available in Appendix 1.

### Procedures

As illustrated in *Figure 1A*, eligible patients received SBRT (administered 5–6 times per week, 3–5 Gy per session) combined with tislelizumab (200 mg on day 1), which was incorporated into the treatment regimen. Each 3-week cycle comprised a maximum of 12 cycles of induction therapy. Patients achieving complete response (CR), PR, or SD transitioned to tislelizumab maintenance therapy (200 mg on day 1) until documented PD, death, unacceptable toxicity, or patient withdrawal of consent. Treatment response was evaluated using CT or MRI after each treatment cycle. Adverse events were systematically monitored and graded according to the National Cancer Institute Common Terminology Criteria for Adverse Events (version 5.0).

The study enrolled 20 eligible patients on November 24, 2022 (*Figure 1B*). All patients received at least one dose of the prescribed regimen. As of the data cutoff date (July 24, 2024), six patients continued to receive maintenance therapy. The median follow-up duration was 15 months (range: 3.4–20.0 months, IQR: 9.6–18.2 months). Due to disease-related complications, specimens could not be obtained from four patients, resulting in 16 patients being included in the per-protocol set.

### Outcomes

The primary endpoints of the study were objective response rate (ORR) and safety, encompassing adverse events and serious adverse events, assessed according to Response Evaluation Criteria in Solid Tumors (RECIST) version 1.1. Secondary endpoints included DCR and PFS. ORR was defined as the proportion of patients who achieved a best objective response of CR or PR per RECIST criteria (version 1.1). DCR was defined as the proportion of patients who achieved CR, PR, or SD according to RECIST criteria (version 1.1). PFS was defined as the time from enrollment to the first documented PD per RECIST version 1.1 or to death from any cause, whichever occurred first.

### CD8, PD-L1, and CD31 expression level

Tumoral CD8 and CD31 expression was measured by immunohistochemistry (IHC) (22C3 pharmDx assays). The sections were scored for staining intensity according to the following scale: 0 (no staining), 1 (weak staining, light yellow), 2 (moderate staining, yellowish brown), and 3 (strong staining, brown), with 0 and 1 considered low expression, and 2 and 3 considered high expression. The score is divided into four levels according to the percentage of positive cells: 0%≤positive cell percentage≤25%, 1 point; 25%<positive cell percentage≤50%, 2 points; 50%<positive cell percentage≤75%, 3 points; 75%<positive cell percentage≤100%, 4 points. IHC score = cell staining intensity score × positive cell percentage score. The PD-L1 CPS was defined as the number of PD-L1 positive cells (tumor cells, lymphocytes, macrophages) as a proportion of the total number of tumor cells multiplied by 100. Positive PD-L1 expression was considered when the CPS was >1.

### NanoString panel RNA sequencing

Due to disease-related limitations, specimens could not be obtained from 4 patients, resulting in a cohort of 16 patients for combined analysis. Tumor tissue samples were collected both before treatment and after treatment. Gene expression of each sample was measured using the NanoString

nCounter platform (NanoString Technologies, Seattle, WA, USA). The quantitative transcriptome data were obtained based on the 289-immuno-gene panel, which includes 289 genes related to the tumor, tumor microenvironment, and immune responses in cancer. The samples that passed the quality control (QC), which included Imaging QC, Binding Density QC, Positive Control Linearity QC, and Positive Normalization QC, can be processed in further analysis. Statistical analyses were conducted using R (version 3.6.1). The raw count data of 289 genes were normalized using the R package NanoStringNorm according to the geometric mean of five housekeeping genes. The log2 transformation was then performed on the normalized data. Differentially expressed genes were identified using the 'DEseq2' package, employing criteria of log2|fold change|>1 and false discovery rate <0.05. Heatmaps depicting the expression patterns of these differentially expressed genes were generated using the 'ComplexHeatmap' package.

## Cox proportional hazards model

Statistical analyses were conducted using R (version 3.6.1). The R package 'survival' was used to generate univariate and multivariate COX analysis results, and the relevant data were saved in *Supplementary file 1*.

## Immune cell profile analyses and additional immune signatures analysis

The determination of immune cell types and gene sets associated with immunotherapy response was informed by established literature sources (*He et al., 2018*; *Zeng et al., 2023*). We transformed each attribute (immune signature or gene set) value (GSVA score) $x_i$ into $x_i'$ by the equation $x_i' = (x_i - x_{min})/(x_{max} - x_{min})$, where $x_{min}$ and $x_{max}$ represent the minimum and maximum of the ssGSEA scores for the gene set across all samples, respectively. The detailed gene signature list can be found in *Supplementary file 2*.

## Gene set enrichment and pathway analysis

The KEGG/GO enrichment analysis was performed using the clusterProfiler R package. The list of gene IDs was used as the input file. The Benjamini-Hochberg method was employed to adjust the p-values. The cutoff threshold of p-values was set to 0.05. The enrichment results were visualized using the ggplot2 R package. The enrichment statistic was set to classic.

## Statistical analyses

PFS and OS were estimated using the Kaplan-Meier method. Statistical analyses were conducted using R (version 3.6.1). Differences between subgroups in terms of efficacy response were assessed using the nonparametric Wilcoxon rank-sum test (Mann-Whitney U test), while comparisons between pre- and posttreatment samples were analyzed using the Wilcoxon signed-rank test. Confidence intervals for response rates were calculated employing the Clopper-Pearson method, with all reported p-values being two-sided. A p-value<0.05 was considered statistically significant.

# Acknowledgements

This research was supported by the Clinical Frontier Technology Program of the First Affiliated Hospital of Jinan University (No. JNU1AF-CFTP-2022-a01223), the National Natural Science Foundation of China (82204436), Natural Science Foundation of Guangdong Province (2024A1515030010, 2022A1515011695), Science and Technology Projects in Guangzhou (2024A03J0825).

# Additional information

## Funding

| Funder | Grant reference number | Author |
|--------|------------------------|--------|
| Clinical Frontier Technology Program of the First Affiliated Hospital of Jinan University | No. JNU1AF-CFTP-2022-a01223 | Yiran Zhang<br>Hanyang Guan<br>Shijin Liu<br>Haoquan Li<br>Zili Bian<br>Jiashuai He<br>Zhan Zhao<br>Shenghui Qiu<br>Tianmu Mo<br>Xiangwei Zhang<br>Zuyang Chen<br>Hui Ding<br>Xiaoxu Zhao<br>Liang Wang<br>Yunlong Pan<br>Jinghua Pan |
| National Natural Science Foundation of China | 82204436 | Yiran Zhang<br>Hanyang Guan<br>Shijin Liu<br>Haoquan Li<br>Zili Bian<br>Jiashuai He<br>Zhan Zhao<br>Shenghui Qiu<br>Tianmu Mo<br>Xiangwei Zhang<br>Zuyang Chen<br>Hui Ding<br>Xiaoxu Zhao<br>Liang Wang<br>Yunlong Pan<br>Jinghua Pan |
| Natural Science Foundation of Guangdong Province | 2024A1515030010 | Yiran Zhang<br>Hanyang Guan<br>Shijin Liu<br>Haoquan Li<br>Zili Bian<br>Jiashuai He<br>Zhan Zhao<br>Shenghui Qiu<br>Tianmu Mo<br>Xiangwei Zhang<br>Zuyang Chen<br>Hui Ding<br>Xiaoxu Zhao<br>Liang Wang<br>Yunlong Pan<br>Jinghua Pan |
| Natural Science Foundation of Guangdong Province | 2022a1515011695 | Yiran Zhang<br>Hanyang Guan<br>Shijin Liu<br>Haoquan Li<br>Zili Bian<br>Jiashuai He<br>Zhan Zhao<br>Shenghui Qiu<br>Tianmu Mo<br>Xiangwei Zhang<br>Zuyang Chen<br>Hui Ding<br>Xiaoxu Zhao<br>Liang Wang<br>Yunlong Pan<br>Jinghua Pan |

| Funder | Grant reference number | Author |
|---|---|---|
| Science and Technology Projects of Guangzhou | 2024A03J0825 | Yiran Zhang<br>Hanyang Guan<br>Shijin Liu<br>Haoquan Li<br>Zili Bian<br>Jiashuai He<br>Zhan Zhao<br>Shenghui Qiu<br>Tianmu Mo<br>Xiangwei Zhang<br>Zuyang Chen<br>Hui Ding<br>Xiaoxu Zhao<br>Liang Wang<br>Yunlong Pan<br>Jinghua Pan |

The funders had no role in study design, data collection and interpretation, or the decision to submit the work for publication.

## Author contributions

Yiran Zhang, Resources, Data curation, Writing - original draft; Hanyang Guan, Validation, Writing - original draft, Writing – review and editing; Shijin Liu, Resources, Investigation, Writing - original draft; Haoquan Li, Investigation, Visualization; Zili Bian, Data curation, Investigation; Jiashuai He, Software, Methodology; Zhan Zhao, Formal analysis; Shenghui Qiu, Validation; Tianmu Mo, Xiangwei Zhang, Data curation; Zuyang Chen, Hui Ding, Xiaoxu Zhao, Liang Wang, Investigation; Yunlong Pan, Jinghua Pan, Conceptualization, Supervision, Project administration, Writing – review and editing

## Author ORCIDs

Hanyang Guan ⓘ https://orcid.org/0009-0002-8853-6151
Yunlong Pan ⓘ http://orcid.org/0000-0002-7434-6560
Jinghua Pan ⓘ https://orcid.org/0000-0003-3741-3397

## Ethics

Clinical trial registration ChicTR2200011777.

This trail was conducted in accordance with the Declaration of Helsinki after approval by the Institutional Review Board of The First Affiliated Hospital of Jinan University (KY-2022-236). All patients provided written informed consent. The ClinicalTrials.gov identifier was: ChiCTR2200066117.

Reviewer #1 (Public review): https://doi.org/10.7554/eLife.103559.3.sa1
Reviewer #2 (Public review): https://doi.org/10.7554/eLife.103559.3.sa2
Author response https://doi.org/10.7554/eLife.103559.3.sa3

# Additional files

## Supplementary files

MDAR checklist

Supplementary file 1. COX analysis results.

Supplementary file 2. Gene signature list.

Source data 1. Nanostring panel RNA sequencing data.

## Data availability

All data generated or analysed during this study are included in the manuscript and supporting files; source data files have been provided for all Figures.

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

# Appendix 1

## Participants

Inclusion criteria included the presence of at least one measurable lesion assessed according to the RECIST version 1.1 and an Eastern Cooperative Oncology Group (ECOG) performance status of 0 or 1. Patients must have received first-line treatment with fluorouracil, irinotecan, and oxaliplatin (any combination) or have been intolerant to first-line drugs. KPS≥60, PS≤2. Patients were also required to have an expected survival of ≥3 months and adequate organ and bone marrow function: absolute neutrophil count >1000/l, hemoglobin >75 g/l, neutrophils ≥1000/µl, blood cells ≥75,000/µl, platelet count >100 × 109/l, prothrombin time <1.5 upper limit of normal (ULN), activated partial thromboplastin time <1.5 ULN, bilirubin ≤1.5×ULN (could be extended to 3×ULN in case of liver metastasis), blood aspartate, aminotransferase, and alanine aminotransferase ≤3×ULN (could be extended to 5×ULN in case of liver metastasis), serum creatinine level ≤1.58×ULN or creatinine clearance ≥40 ml/min, urinary protein/creatinine ratio < 1 (or urine analysis < 1 + or 24 hr urinary protein <1 g/24 hr). Patients were excluded in the following cases: received previous treatment with any radiation therapy; received treatment with corticosteroids or other immunosuppressive agents within 14 days prior to study drug administration, presence of autoimmune disease, known interstitial lung disease, and those with previous or concurrent malignancies except for basal cell carcinoma, cutaneous squamous cell carcinoma, or cervical carcinoma in situ that have undergone radical treatment.

A list of 289 genes of NanoString panel RNA sequencing.

| SDHA | ITGAL | TNFSF13B | CTSW | PTGER4 | GNLY |
|------|-------|----------|------|--------|------|
| C1QB | PSMB10 | HSD11B1 | IRF1 | CXCR2 | PDCD1LG2 |
| BTLA | C1QA | GZMA | TDO2 | HERC6 | ABCF1 |
| TGFB1 | LYZ | ITGA1 | CD28 | IL17A | IL6 |
| GUSB | CD274 | CXCR3 | AXL | IL15 | FPR1 |
| TNFSF9 | IL4 | PDCD1 | IFI35 | CD3D | CD74 |
| ICOS | KIR2DL3 | SELL | IL10RA | CD1C | G6PD |
| PECAM1 | IL7R | BRCA1 | CCR5 | EOMES | EGFR |
| SPIB | IL21R | CD40 | TFRC | TLR7 | HLA-DMB |
| IFIH1 | IL2RG | IFIT2 | ENTPD1 | LY9 | HLA-DPA1 |
| MTOR | FCGR2B | HLA-DQA2 | ITGAE | HAVCR2 | IRF4 |
| TRAT1 | CD70 | NOS2 | LCK | IFNG | CDKN2A |
| DLL4 | CD3E | CD79A | CCR2 | FCRL2 | EIF2AK2 |
| NFKBIA | RB1 | CD68 | KIR3DL2 | MS4A2 | CTAG1B |
| ADM | HLA-DMA | GZMB | CCL20 | CSF3R | MS4A1 |
| HLA-DOB | S100A9 | CCL18 | MAGEA12 | CXCL12 | NT5E |
| CXCL11 | CYBB | CD86 | TNF | IFIT1 | PTPRC |
| CCR4 | CPA3 | FGF13 | FUT4 | ICOSLG | IL2 |
| CD40LG | CD209 | CCL22 | IFI27 | VEGFA | BRCA2 |
| CD163 | NKG7 | CCL2 | STAT4 | TLR9 | IL12RB2 |
| PRF1 | CD44 | IL2RB | ITGAM | CXCL9 | TBP |
| CD244 | CXCR4 | PTEN | TNFRSF9 | MRC1 | CXCL1 |
| GZMH | SNAI1 | MS4A4A | TCL1A | TNFSF18 | IDO1 |
| CD247 | VTCN1 | MKI67 | CCL21 | TIGIT | ATM |
| CD48 | PSMB9 | LILRB2 | CSF2 | PDGFB | IL18 |

*Continued on next page*

*Continued*

| SDHA | ITGAL | TNFSF13B | CTSW | PTGER4 | GNLY |
|---|---|---|---|---|---|
| IL10 | TNFSF10 | SIGLEC5 | VCAM1 | HDC | MAGEA4 |
| ANGPT2 | PMS2 | TAP1 | RAD51 | CD276 | S100A8 |
| CD80 | CCL13 | BIRC5 | CD3G | ZAP70 | ISG15 |
| ZEB1 | CTSS | TNFRSF14 | TNFRSF18 | PVR | MMP9 |
| CD69 | PIK3CA | TNFRSF4 | PTGS2 | BLK | IL1A |
| CEACAM3 | CMKLR1 | ARG1 | S100A12 | OAS3 | TWIST1 |
| CD84 | PNOC | TNFRSF17 | IL1B | BLM | CCL5 |
| KLRD1 | CD8A | PIK3CD | RUNX3 | CSF1R | ADORA2A |
| IFIT3 | CCL4 | HLA-DOA | CX3CR1 | IFI6 | BRIP1 |
| CD47 | CCND1 | TNFRSF1B | IRF9 | SH2D1A | MAGEA1 |
| CXCL8 | IFITM2 | CD38 | CD27 | NECTIN2 | MLANA |
| ITGB2 | TLR3 | CXCL5 | STAT3 | AKT1 | ITGAX |
| CTLA4 | MAGEC2 | PDGFA | FCGR1A | MLH1 | KLRK1 |
| TBX21 | TNFRSF1A | FOXP3 | MYC | TIE1 | SLAMF7 |
| MELK | GBP1 | CD8B | CD79B | CD14 | CXCL13 |
| MX1 | TNFSF4 | CSF2RB | RORC | FCAR | CCL7 |
| KLRB1 | PTPN11 | BCL2 | CD2 | HLA-DRA | LAG3 |
| MSH2 | KIR3DL1 | TLR8 | CD4 | CXCL10 | ICAM1 |
| CX3CL1 | FASLG | OAS2 | CXCR6 | HIF1A | NCAM1 |
| GZMK | POLR2A | NCR1 | NBN | OAS1 | IFITM1 |
| CD19 | FAS | CD6 | STAT1 | MSH6 | IL2RA |
| TPSAB1 | HLA-DQA1 | PRR5 | GZMM | HLA-E | PF4 |
| HLA-DRB1 | UBB | OAZ1 | TBC1D10B | FCGR3B | CXCL2 |
| STK11IP | | | | | |

