## [Editor Report · eLife Assessment]

This **valuable** study found Gamma Knife SBRT combined with tislelizumab offers a safe and powerful later-line option for pMMR/MSS/MSI-L metastatic CRC patients who were unresponsive to the first and second-line chemotherapy. The authors implemented a well-structured experimental protocol and provide **convincing** evidence to substantiate the conclusions. This work would be of broad interest to oncologists working on colorectal cancer.

---

## [Referee Report · Reviewer #1 (Public review)]

Summary:

This study presents compelling evidence for a novel treatment approach in a challenging patient population with MSS/pMMR mCRC, where traditional immunotherapy has often fallen short. The combination of SBRT and tislelizumab not only yielded a high disease control rate but also indicated significant improvements in the tumor's immune landscape. The safety profile appears favorable, which is crucial for patients who have already undergone multiple lines of therapy.

Strengths:

The results underscore the potential of leveraging radiation therapy to enhance the effectiveness of immunotherapy, especially in tumor environments previously deemed hostile to immune interventions. Future research should focus on larger cohorts to validate these findings and explore the underlying mechanisms of immune modulation post-treatment.

Comments on revisions:

The author provided satisfactory responses to my queries, offering clarifications and additional explanations to address potential points of confusion. The supplementary experimental data further corroborate the author's conclusions. Although a more in-depth and detailed analysis did not yield significant results, this does not undermine the overall integrity of the article's structure or the reliability of its conclusions. Based on the content and the supporting evidence presented, I believe this article meets the necessary criteria for publication.

---

## [Referee Report · Reviewer #2 (Public review)]

Summary:

This Phase II clinical trial investigates the combination of Gamma Knife Stereotactic Body Radiation Therapy (SBRT) with Tislelizumab for the treatment of metastatic colorectal cancer (mCRC) in patients with proficient mismatch repair (pMMR). The study addresses a critical clinical challenge in the management of pMMR CRC, focusing on the selection of appropriate candidates. The results suggest that the combination of Gamma Knife SBRT and Tislelizumab provides a safe and potent treatment option for patients with pMMR/MSS/MSI-L mCRC who have become refractory to first- and second-line chemotherapy. The study design is rigorous, and the outcomes are promising.

Advantage:

The trial design was meticulously structured, and appropriate statistical methods were employed to rigorously analyze the results. Bioinformatics approaches were utilized to further elucidate alterations in the patient's tumor microenvironment and to explore the underlying factors contributing to the observed differences in treatment efficacy. The conclusions drawn from this trial offer valuable insights for managing advanced colorectal cancer in patients who have not responded to first- and second-line therapies.

Weakness:

(1) Clarity and Structure of the Abstract

- Results Section: The results section should contain important data, I suggest some important sequencing data should be shown to enhance understanding.

(2) As the author using the NanoString assay for transcriptome analysis, more detail should be shown such as the version of R, and the bioinformatics analysis methods.

(3) It is interesting for included patients that PD-L1 increase expression after Gamma Knife Stereotactic Body Radiation Therapy (SBRT) treatment, How to explain it?

(4) It would be helpful to include a brief discussion of the limitations of the study, such as sample size constraints and their impact on the generalizability of the results. This will give readers a more comprehensive understanding of the findings.

(5) Language Accuracy: There are a few instances where wording could be more professional or precise.

Revision comment:

The author had responded to all questions and improved the manuscript. The author's answers and revisions are very satisfactory to me. I believe it is an important study for the immunotherapy of colorectal cancer.

---

## [Author Response]

The following is the authors’ response to the original reviews

**Public Reviews:**

**Reviewer #1 (Public review):**
Summary:This study presents compelling evidence for a novel treatment approach in a challenging patient population with MSS/pMMR mCRC, where traditional immunotherapy has often fallen short. The combination of SBRT and tislelizumab not only yielded a high disease control rate but also indicated significant improvements in the tumor's immune landscape. The safety profile appears favorable, which is crucial for patients who have already undergone multiple lines of therapy.Strengths:The results underscore the potential of leveraging radiation therapy to enhance the effectiveness of immunotherapy, especially in tumor environments previously deemed hostile to immune interventions. Future research should focus on larger cohorts to validate these findings and explore the underlying mechanisms of immune modulation post-treatment.Weaknesses:I believe the author's work is commendable and should be considered with some minor modifications:(1) While the author categorized patients based on the type of RAS mutation and the location of colorectal cancer metastasis, the article does not adequately address how these classifications influence treatment outcomes. Such as whether KRAS or NRAS mutations, as well as the type of metastatic lesions, affect the sensitivity to gamma-ray treatment and lead to varying responses.

Thank you very much for your question. Therefore, in the revised manuscript, we added an analysis of the impact of RAS mutation types and different metastatic sites on patient prognosis, but unfortunately, due to the limited number of samples, we were unable to obtain satisfactory results. We also placed the relevant results in the supplementary figure.

(2) In Figure 2, clarification is needed on how the author differentiated between on-target and off-target lesions. I observed that some images depicted both lesion types at the same level, which could lead to confusion.

We sincerely apologize for any oversight in our previous submission. To clarify, during the process of radiotherapy planning, we pre-select target lesions at the CT image level, and subsequently define the planning treatment volume (PTV) by marking these pre-selected areas with the 50% isodose lines. In our efficacy evaluation, we distinguish between the target lesions inside the PTV and any lesions outside the target area. In response to your valuable feedback, we have now added the isodose lines for the target lesions to the supplementary figure for greater clarity.

(3) The author performed only a basic difference analysis. A more comprehensive analysis, including calculations of markers related to treatment efficacy, could offer additional insights for clinical practice.

To identify potential markers associated with treatment efficacy, we attempted to establish a Cox proportional hazards model and conducted both univariate and multivariate Cox regression analyses. Unfortunately, due to the constraints of sample size and sequencing depth, the analyses did not yield statistically significant results, and we were unable to identify markers that could clearly predict treatment outcomes.

(4) The transcriptome sequencing analysis provides insights into how stereotactic radiotherapy sensitizes immunotherapy; however, it currently relies on a simple pre- and post-treatment group comparison. It would be beneficial to include additional subgroups to explore more nuanced findings.

We acknowledge the limitations in the depth of our analysis. In addition to performing differential analysis between the responder group (PR) and the non-responder group (Non-PR), we also conducted differential gene expression analysis on samples before and after treatment. The results revealed a consistent increase in the expression of NOS2 in both groups following Gamma Knife combined with immunotherapy, suggesting that this gene may serve as a potential prognostic factor influencing treatment outcomes. However, given the limited number of studies exploring the role of NOS2 in this context, we recognize that further research is necessary to better understand its involvement and to substantiate its potential as a predictive marker.

(5) The author briefly discusses the effects of changes in tumor fibrosis and angiogenesis on treatment outcomes. Further experiments may be necessary to validate these findings and investigate the underlying mechanisms of immune regulation following treatment.

We sincerely appreciate your thoughtful feedback on our results. In response, we conducted additional experiments, including immunohistochemical analysis of patient samples before and after combined treatment. The results demonstrated a reduction in the expression of CD31, a marker of tumor angiogenesis, following the combined treatment. This finding further supports our hypothesis that Gamma Knife treatment, in combination with immunotherapy, may effectively inhibit tumor angiogenesis, contributing to an improved therapeutic outcome.

**Reviewer #2 (Public review):**
Summary:This Phase II clinical trial investigates the combination of Gamma Knife Stereotactic Body Radiation Therapy (SBRT) with Tislelizumab for the treatment of metastatic colorectal cancer (mCRC) in patients with proficient mismatch repair (pMMR). The study addresses a critical clinical challenge in the management of pMMR CRC, focusing on the selection of appropriate candidates. The results suggest that the combination of Gamma Knife SBRT and Tislelizumab provides a safe and potent treatment option for patients with pMMR/MSS/MSI-L mCRC who have become refractory to first- and second-line chemotherapy. The study design is rigorous, and the outcomes are promising.Advantage:The trial design was meticulously structured, and appropriate statistical methods were employed to rigorously analyze the results. Bioinformatics approaches were utilized to further elucidate alterations in the patient's tumor microenvironment and to explore the underlying factors contributing to the observed differences in treatment efficacy. The conclusions drawn from this trial offer valuable insights for managing advanced colorectal cancer in patients who have not responded to first- and second-line therapies.Weakness:(1) Clarity and Structure of the Abstract- Results Section: The results section should contain important data, I suggest some important sequencing data should be shown to enhance understanding.

Thank you for your insightful question. In response, we have revised the content of the article and restructured the abstract to enhance its scientific clarity and make it more accessible to readers.

(2) As the author using the NanoString assay for transcriptome analysis, more detail should be shown such as the version of R, and the bioinformatics analysis methods.

We have also addressed the missing details in our research methodology. The revised manuscript now includes a complete description of the research methods, along with the specific software and versions used.

(3) It is interesting for included patients that PD-L1 increase expression after Gamma Knife Stereotactic Body Radiation Therapy (SBRT) treatment, How to explain it?

Thank you for your thought-provoking question. PD-L1 plays a crucial role in tumor cell immune evasion, and anti-PD-1/PD-L1 inhibitors have emerged as effective immune checkpoint inhibitors, widely used in cancer therapy. In our clinical trials, we observed an increase in PD-L1 expression in some patients following combined treatment. Existing literature suggests that activation of various carcinogenic and stress response pathways, along with post-transcriptional modifications of PD-L1 (such as phosphorylation, glycosylation, acetylation, ubiquitination, and palmitoylation), can influence its expression[1]. We hypothesize that the increase in PD-L1 expression may be attributed to the activation of specific signaling pathways induced by the radiation from Gamma Knife treatment, as well as the enhanced tumor stress in response to the treatment. However, the precise mechanisms underlying this observation require further experimental investigation. A deeper understanding of these processes could potentially optimize our clinical treatment strategies.

(4) It would be helpful to include a brief discussion of the limitations of the study, such as sample size constraints and their impact on the generalizability of the results. This will give readers a more comprehensive understanding of the findings.

Thank you for highlighting the limitations of the article. In response, we have added a detailed discussion of the constraints arising from the limited number of experimental samples and insufficient sequencing depth. This addition aims to provide readers with a clearer understanding of the study's limitations and the context of our research findings.

(5) Language Accuracy: There are a few instances where wording could be more professional or precise.

Regarding the language deficiency, we are very sorry that the wording of the professional content in the article is not careful and accurate enough due to the difference in the native language environment. We have checked our article again and revised the wording and grammar in the hope that you and other readers can grasp our research content more accurately.

**Recommendations for the authors:**

**Reviewer #1 (Recommendations for the authors):**
The research presented in this article is commendable; however, I would like to propose several revisions for consideration:Consideration of Concomitant Medications: It is imperative to ascertain whether enrolled patients utilized additional pharmacological agents alongside the trial regimen. Such concurrent drug use could potentially influence the final outcomes. A concise discussion of this aspect is warranted within the manuscript.Clinical Characterization of Response Groups: An examination of the clinical characteristics distinguishing the effective and non-responsive cohorts within the trial is essential. This inquiry merits further exploration, as it may elucidate factors influencing treatment efficacy.Tumor Microenvironment Analysis: The authors highlight the implications of tumor fibrosis and angiogenesis on therapeutic response. Identification of specific biomarkers associated with these phenotypes is crucial. I recommend undertaking straightforward testing and validation to substantiate these observations.

Thank you very much for your valuable suggestions, many of which have been incorporated into the revised manuscript. Regarding the consideration of concurrent medication, we would like to clarify that all patients included in the study were advanced CRC patients who had progressed during first- or second-line treatments. As such, targeted therapy or chemotherapy was used concurrently in the trial. Previous studies have not indicated that different targeted therapies influence the efficacy of Gamma Knife treatment, though some chemotherapy agents may vary in their side effects. However, we believe these differences do not significantly impact the final outcomes. Given that existing chemotherapy regimens do not substantially affect patient prognosis, we considered the combined drug treatment regimen to be an irrelevant variable in our analysis.

Additionally, we have carefully examined the clinical characteristics of patients across different groups. We have also included an analysis of the impact of various mutation types and metastatic sites in the revised manuscript. Furthermore, we plan to perform CD31 staining on lesions from both the responder and non-responder groups before and after Gamma Knife treatment to assess the role of angiogenesis in treatment response.

**Reviewer #2 (Recommendations for the authors):**
The abstract should be revised for greater clarity and include key results that substantiate the conclusions. The discussion section needs to more thoroughly address the limitations of the clinical trial, providing readers with a deeper understanding of the trial's findings and implications. Additionally, the methods section should be more rigorous and detailed, offering sufficient information to enhance the transparency and robustness of the experimental design.

Thank you for your constructive suggestions regarding the shortcomings in our manuscript. In response, we have thoroughly reviewed the article and addressed the missing content, including revisions to the abstract, results, discussion, and methods sections. Additionally, we have refined the grammar and wording throughout the manuscript to enhance its professionalism and ensure it aligns with the standards expected for publication.

(1) YAMAGUCHI H, HSU J M, YANG W H, et al. Mechanisms regulating PD-L1 expression in cancers and associated opportunities for novel small-molecule therapeutics [J]. Nature reviews Clinical oncology, 2022, 19(5): 287-305.